# Impact of Dietary Fiber on Inflammation and Insulin Resistance in Older Patients: A Narrative Review

**DOI:** 10.3390/nu15102365

**Published:** 2023-05-18

**Authors:** Michele Niero, Giulio Bartoli, Paolo De Colle, Marialaura Scarcella, Michela Zanetti

**Affiliations:** 1Geriatric Clinic, Maggiore Hospital, Azienda Sanitaria Universitaria Giuliano Isontina, 34148 Trieste, Italy; niero.michele92@gmail.com (M.N.); giulio.bartoli@asugi.sanita.fvg.it (G.B.); paolo.decolle@asugi.sanita.fvg.it (P.D.C.); 2Department of Medical Sciences, University of Trieste, 34127 Trieste, Italy; 3Anesthesia, Intensive Care and Nutritional Science, Azienda Ospedaliera “Santa Maria”, Via Tristano di Joannuccio, 05100 Terni, Italy; laura.scarcella@yahoo.it

**Keywords:** dietary fiber, inflammaging, insulin resistance, older adult, acute disease

## Abstract

The beneficial impact of dietary fiber on the prevention and management of several chronic conditions associated with aging, including diabetes, neurodegenerative, cardiovascular diseases, and cancer, is well-known. High fiber intake has been associated with reduced inflammatory mediators counteracting the low-grade chronic inflammation typical of older age. In addition, dietary fiber improves postprandial glucose response and insulin resistance. In contrast, during acute diseases, its effects on insulin resistance and modulation of immune response are unclear. The aim of this narrative is to summarize the evidence for the potential impact of dietary fiber on inflammation and insulin resistance in older adults, with a particular focus on those acutely ill. Available evidence suggests that dietary fiber has the potential to counteract acute inflammation and to improve metabolic health. In addition, modulation of gut microbiota composition may contribute to improved immune function, particularly in the setting of aging-associated dysbiosis. This phenomenon has relevant implications in those acutely ill, in whom dysbiosis can be exacerbated. Our review leads to the conclusion that dietary interventions based on fiber manipulation could exploit its beneficial effects on inflammation and insulin resistance, if conducted from a precision nutrition perspective. This could also be true for the acutely ill patient, even though strong evidence is lacking.

## 1. Introduction

Dietary fiber plays a key role in several aging-associated diseases. Consistent evidence shows a correlation between low dietary fiber intake and increased cardiovascular mortality, cancer incidence, impaired cognitive function, and physical performance. In older adults, higher cereal fiber intake results in the reduction of various inflammatory markers and in a lower risk of cardiovascular disease [1]. This is particularly important because a state of chronic inflammation characterizes the aging process. It is well-established that the modulation and maintenance of the homeostasis of the inflammatory response are reduced by advancing age [2]. Several molecular mechanisms have been described that link aging with persistent low-grade chronic inflammation. Furthermore, it now seems clear that this inflammatory state is, at least in part, at the origin of the development of several diseases associated with aging, including cardiovascular and neurodegenerative diseases, diabetes, metabolic syndrome, immune system disorders, infections, and cancer. This sterile and chronic inflammatory state is known as inflammaging [3,4]. In addition, inflammation is often associated with insulin resistance in chronic diseases, [5] and this combination results in excess risk mortality and averse clinical outcomes. Acute intercurrent diseases will aggravate this situation with relevant consequences in older patients. The potential impact of dietary fibers on clinical outcomes of older patients, particularly those acutely ill, and their effects on inflammation and insulin resistance has not been previously assessed. Therefore, the aim of this narrative review is to give some insights into the relationship between inflammaging and insulin resistance in older adults, and to explore how these conditions can be modified by dietary fiber, with a particular focus on acutely ill elderly patients.

## 2. Mechanisms of Chronic Inflammation in Older Adults

The term “inflammaging” refers to a chronic proinflammatory state typical of old age, characterized by a reduction in the ability to cope with stressful events and by an accumulation of senescent cells, due to a continuous antigenic load and stress and to alterations of the tissue regeneration systems [6,7,8]. In the last two decades, many studies have been published based on this concept, outlining even more precisely the characteristics of this phenomenon and its underlying mechanisms [9]. Furthermore, a causal relationship has been described between the accumulation of senescent cells in organs and the development of several age-related diseases, such as Alzheimer’s disease and related dementias, including vascular contributions to cognitive impairment [10].

Cellular senescence is induced by DNA damage, which may be due to two fundamental mechanisms: the shortening of telomeres at each cell replication cycle and stress-induced premature senescence (SIPS). Telomere shortening is responsible for replicative senescence, which is intended to prevent the development of mutations that could occur after several replicative cycles, becoming dangerous for the whole organism [11]. On the other hand, SIPS is a condition of cellular senescence linked to exposure to damaging factors such as irradiation, inflammation, or oxidative stress, which cause premature cellular aging, independently of telomere length [12]. It can also be favored by metabolic disorders such as obesity and diabetes, conditions defined by premature metabolic aging [13].

Senescent cells undergo phenotypic changes, altered paracrine activity, and increased production of reactive oxygen species, but their most important feature is the secretion of senescence-associated secretory phenotype, a cocktail of growth factors, proteases, chemokines, and proinflammatory cytokines, such as Interleukin (IL)-6, IL-1β, IL-1α, and TGF-β through NFkB [14,15,16,17], whose physiological role would be to attract the cells of the immune system towards the senescent cell to induce apoptosis. However, in geriatric age, impairment of the immune system, a phenomenon known as immunosenescence, leads to the inability to effectively eliminate senescent cells, which then accumulate in several organs and produce a persistent and afinalistic inflammatory state, which spreads by the paracrine way, also damaging the adjacent structures [2,14]. In addition, impaired activity of the NACHT, LRR, and PYD domains-containing protein 3 (NLRP3) inflammasome, a unique enzyme system that induces an inflammatory state in the absence of an overt infection [18], has been demonstrated in the aging population, suggesting its contribution to the low-grade inflammation which characterizes this condition, through the activation of precursors of several proinflammatory cytokines such as IL-1β, IL-1α, and IL-18 [2,18,19].

Moreover, alterations in the composition of gut microbiota and gut permeability, which characterize aging, are among the main sources of low-grade inflammation in the older adult. In fact, it seems that age-related dysbiosis can contribute both to the phenomenon of immunosenescence and to inflammaging. Several associations have been observed between dysbiosis and chronic enteric and extraenteric conditions typical of old age [19].

In summary, several underlying mechanisms contribute to the development of chronic low-grade inflammation or inflammaging typical of older individuals, which have been associated with insulin resistance and chronic diseases. Among these, age-related dysbiosis plays an important role. Due to the possibility to manipulate the composition of intestinal microbiota with nutritional interventions, particularly dietary fiber, this strategy has the potential to foster changes in the microbiome to reverse aging-associated dysbiosis.

The figure summarizes the main mechanisms that induce a proinflammatory phenotype in the elderly patient (Figure 1).

## 3. Inflammaging and Insulin Resistance

Although a clear molecular mechanism has not yet been defined, studies are growing that demonstrate several connections between hyperglycemia and the development of cellular senescence; both phenomena are characterized by a chronic proinflammatory phenotype and they seem to influence mutually [16,18].

The role played by metabolic disorders in inducing a proinflammatory and senescent cell phenotype is known as premature metabolic aging [13]. Wiley et al. suggested that four metabolic factors are involved in accelerated cell senescence: mitochondrial dysfunction, oxygen, disrupted NAD+ metabolism, and hyperglycemia [16]. Prolonged hyperglycemia, as a result of insulin resistance, results in increased ROS formation and reduced antioxidant defenses, leading to increased oxidative stress and a global proinflammatory state [20]. It is interesting to underline that some molecular pathways have been identified which, if overactivated, are able to induce both a low-grade chronic inflammation phenotype, typical of inflammaging, and a state of insulin resistance. This is the case of upregulation of NLRP3 [18] and of Forkhead transcription factor 6 (FoxO6) [21]. Finally, some studies suggest that both diabetes and advanced age induce a shortening of telomeres [22].

Some recent studies also suggest that there may be a link between age-associated decline in mitochondrial function and the onset of insulin resistance. There seems to be, for example, an inverse correlation between the number of mitochondria contained in the muscle cells of the elderly and insulin sensitivity. Probably, the pathogenesis of this phenomenon is linked to the production of ROS by the impaired mitochondria and by the accumulation of intracellular lipids that induce insulin resistance in skeletal muscle [23].

With regard to acute conditions, the onset of stress-related hyperglycemia can often be observed which, if mild to moderate, can be considered a physiological response aimed at guaranteeing greater nutrition to the immune cells [24,25]. However, higher levels of stress-related hyperglycemia may be harmful and worsen the damage related to the acute condition, through increased ROS production and oxidative stress, endothelial, vascular, and immune system dysfunction and the induction of alterations in the inflammatory response [26]. In fact, it has been observed that high blood glucose levels induced by the acute phase stress response are associated with worse prognostic outcomes in hospitalized elderly patients [27]. In 2015, the stress hyperglycemia ratio (SHR) was developed, a tool to identify patients at risk of stress-related hyperglycemia, based on the ratio between blood glucose at hospital admission and the previous average values extrapolated from glycated hemoglobin [28]. A retrospective study conducted on patients hospitalized for sepsis identified SHR as a risk factor for increased in-hospital mortality [29].

Another recent retrospective multicenter Italian study, conducted on 4714 patients over 65 years old hospitalized in Internal Medicine and Geriatrics, demonstrated that admission blood glucose levels values greater than 250 mg/dL were associated with a higher degree of comorbidities assessed using the Modified Cumulative Illness Rating Scale [30] and higher in-hospital mortality, regardless of the presence of diabetes in medical anamnesis and of the admitting diagnosis [31]. Similar results have also been found in studies conducted in other countries [32].

Altogether available evidence suggests a link between insulin resistance and low-grade chronic inflammation. In acute illness, insulin resistance contributes to the development of stress hyperglycemia, which is often associated with inflammation; hyperglycemia-stimulated ROS production results in increased production of proinflammatory cytokines [33], suggesting a close link between metabolism and immunity.

## 4. Inflammation in the Older Adult Acutely Ill

Aging is associated with an increased rate of hospitalization due to acute illnesses, as well as an increase in mortality and morbidity rates [34].

While inflammaging represents an alteration of the inflammatory process that produces progressive tissue damage, acute inflammation is a physiological phenomenon aimed at counteracting damage to a specific region of the body and enabling the rest of the immune system to take action to eliminate the damaging factor [35].

Not much is known about the role that inflammaging plays in the acute phase of disease; however, a high grade of baseline systemic inflammation is known to correlate with worse clinical outcomes in patients with acute infections [36]. It has also been shown that elevated plasma levels of proinflammatory cytokines, such as interleukin 6, and low levels of albumin correlate with higher mortality in hospitalized older patients regardless of the degree of frailty [37]. The Glasgow Prognostic Score, a simple tool based on the measurement of albumin and C-reactive protein levels, is a proxy of systemic inflammation of patients acutely ill [38]. It predicts mortality in many clinical settings such as cancer, but also in COPD, relapse of idiopathic pulmonary fibrosis, and myocardial infarction in the elderly [39,40,41,42]. Therefore, it could be hypothesized that a condition of systemic inflammation, such as inflammaging, is associated with a lower ability of the elderly patient to cope physiologically with an acute event, and therefore with greater mortality and morbidity.

Following the SARS-CoV-2 pandemic, several articles have been published which have tried to outline the peculiarities of the manifestations of an acute disease such as COVID-19 in the older adult, also trying to explain the higher mortality that this condition determined in old age and in patients with age-related comorbidities, such as diabetes and cardiovascular disease, and the lower efficacy of vaccines in this segment of the population. Immunosenescence and inflammaging appear to be risk factors for the development of more severe forms of COVID-19 disease [43]. In the first place, inflammaging is at the basis of the development of several disorders typical of old age, such as diabetes and atherosclerosis, which overall make the homeostatic balance of the organism more fragile, resulting in worse response outcomes to an acute disease [44,45]. Furthermore, it was observed that an immunological profile with high levels of proinflammatory cytokines was associated with severe forms of COVID-19 [43]. In inflammaging, then, the molecular mechanisms that normally downregulate the inflammatory process, such as the antiinflammatory cytokine families of IL-10 and TGF-β, are less represented, making it more difficult to overcome the acute phase of the inflammatory cascade and to start the repair processes [43].

Furthermore, the NLRP3 inflammasome pathway, which, as seen previously, is already upregulated in inflammaging, is also activated by SARS-CoV-2 endosomal replication, as well as by characteristics more generally common to acute infections, such as oxidative stress, DNA damage, necrotic cell damage, and production of multiple DAMPs. Excess activation of the NLRP3 inflammasome contributes to the development of the so-called “cytokine storm” [46,47]. It has been observed that one of the main risk factors for the development of acute respiratory distress syndrome (ARDS) in COVID-19 is advanced age [48]. In fact, older adults are more likely to experience severe acute respiratory failure with impaired gas exchange during COVID-19 due to an underlying state of increased airway inflammation and fibrosis. For the same reason, elderly patients find it more difficult to overcome COVID-19 illness and respond to ICU ventilation [43]. The decline in number and function of alveolar macrophages, which is characteristic of inflammaging, may also play a role in delaying the onset of an effective inflammatory response in the elderly patient with COVID-19 [43]. Furthermore, it seems that the alteration of the balance between alveolar macrophages, NK, and CD8 T lymphocytes, observed in patients with chronic inflammation, leads to an excessive increase in T cells during the acute phase of SARS-CoV-2 infection promoting fibrosis and reduced lung function [49].

Another molecular mechanism common to inflammaging and acute inflammation is the reduction of the activity of Nicotinamide Adenine Dinucleotide (NAD), an important coenzyme in energy metabolism, that acts as an electron transfer agent in numerous chemical redox reactions [50]. The expression of CD38 (an enzyme responsible for the elimination of NAD) increases during inflammation [51]. A recent study conducted on mice macrophages demonstrated that both inflammaging and the acute phase of inflammation induce an accumulation, especially in the white visceral adipose tissue and liver, of proinflammatory M1-like macrophages, responsible for producing large amounts of CD38, thereby reducing NAD levels in these tissues. It seems that senescent cells, through SASPs, induce macrophages to proliferate and express CD38 [52].

All these mechanisms and available evidence explain the worse prognosis of elderly patients with a high degree of inflammaging in the face of the acute phases of a disease.

## 5. Fiber, Inflammation, and Insulin Resistance

The role of dietary fiber in reducing systemic and intestinal inflammation is well-demonstrated. This effect has been widely shown for soluble fibers and is attributable to their beneficial effects on shaping the intestinal microbiota and on the production of metabolites (short-chain fatty acids, SCFA) with pleiotropic effects, including those that are antiinflammatory [53]. In vitro studies have demonstrated that in vitro soluble fiber inhibits the NF-κB signaling pathway, resulting in the reduced production of proinflammatory mediators such as COX2 [53]. Matt and colleagues demonstrated that a higher fiber diet is associated with increased expression of butyrate in mice, which is associated, especially in aged mice, with a reduction in the expression of proinflammatory cytokines in microglia, thus delaying brain ageing [54]. In general, soluble fiber demonstrates favorable metabolic effects, which include improved insulin resistance and glucose tolerance that are associated with reduced cholesterol levels and blunted systemic inflammatory response [55]. In addition, a recent systematic literature review, including 31 randomized clinical trials, showed that intake of foods containing whole grains has beneficial effects on systemic levels of inflammatory markers [56]. These findings were recently confirmed also in older adults by Shivakoti R et al. [1], who showed in a large cohort of 4125 individuals that by increasing cereal fiber intake by 5 g/day, a significant reduction of C-reactive protein (CRP) and of interleukin 1 receptor antagonist could be demonstrated. These changes were associated with a concomitant decrease in the incidence of cardiovascular disease. The authors also showed that inflammation was responsible for about one-sixth of the association between dietary fiber and cardiovascular events. Unfortunately, the effects on glucose metabolism in this cohort were not assessed. Further evidence confirms the antiinflammatory potential of dietary fiber in vivo in older adults. Vulevic et al. demonstrated that administration of a prebiotic mixture containing galactooligosaccharides to a cohort of healthy elderly subjects resulted in reduced production of proinflammatory cytokines IL-6, IL1β, and tumor necrosis α, and in increased levels of the antiinflammatory cytokine IL-10 [57]. Similar findings were obtained by Guigoz et al., who showed that administration of a prebiotic containing fructooligosaccharides to malnourished frail older residents of a nursing home resulted in reduced expression of IL-6 and tumor necrosis α mRNA [58]. Taken together, these findings suggest that dietary fiber exerts an antiinflammatory effect in elderly persons.

Previous studies on fiber consumption and glucose metabolism in the general population have demonstrated beneficial effects, possibly mediated by several mechanisms, including delayed substrate absorption, increased glucose oxidation, and reduced release of free fatty acids [59]. Moreover, high-fiber diets may improve insulin sensitivity by reducing hepatic liver content [60]. Following acute fiber ingestion, a blunted glucose and insulin response has been described [61]. A recent study in patients with a previous episode of acute pancreatitis demonstrated that, in those diagnosed with prediabetes or diabetes after the acute event, higher dietary fiber intake was associated with both lower fasting glucose and lower glycated hemoglobin [62]. While dietary fiber is associated with improved metabolic control in nonhospitalized individuals [63], there is limited evidence on the impact in acutely ill patients. This is particularly important in older adults characterized by chronic-low grade inflammation. A prospective study conducted in a cohort of older (71.8 years) neurocritical care patients who received an enteral formula containing fructooligosaccharides or a standard formula demonstrated that the group fed the prebiotic-containing formula had a lower insulin requirement and reduced CRP levels [64]. Although stress-induced hyperglycemia is common in intensive care unit patients who are typically younger than those admitted in other settings of care, insulin resistance and hyperglycemia are common also outside the intensive care unit insulin among older nondiabetic patients acutely hospitalized, and is associated with poor clinical outcomes [65,66,67]. Whether dietary fiber intake may improve metabolic control and inflammatory markers in these patients is currently unknown. This information has potentially important consequences on clinical management, since a targeted nutrition intervention based on fiber supplementation may improve outcomes. Indeed, the metabolic milieu that develops following an acute event in nondiabetic patients admitted to the hospital is a complex state characterized by increased serum levels of proinflammatory cytokines and of counter-regulatory hormones that promote gluconeogenesis and insulin resistance [68]. Regarding safety issues, the literature suggests that the risk of adverse events due to fiber administration is low [69]. In older adults with acute diseases, possible interactions of dietary fiber with drug absorption should be considered before their administration. Older studies suggested that the intestinal absorption of the antidiabetic drug glibenclamide was impaired by the concomitant ingestion of glucomannan [70]; in addition, levothyroxine absorption might be slightly, although not significantly, reduced by dietary fiber [71], as well as digoxin [72]; in the latter case, drug formulation (capsule versus tablet) may play an important role. Moreover, amoxicillin bioavailability was significantly lower in healthy volunteers fed a high-fiber diet for three days as compared to the control group fed a standard diet [73]. It should be noted that not only negative fiber-drug interactions have been reported with regard to intestinal absorption; in some cases prebiotics demonstrate an enhancing effect on intestinal assimilation, although timing of consumption, type of prebiotic, and characteristics of the population seem to play an important role [74]. Examples of these prebiotics include levodopa [75] and iron [74].

To summarize, the available literature suggests that dietary fiber has the potential to modulate inflammation in older adults with chronic conditions. In the acute setting, evidences are scarce, although clinical benefits might be relevant, once potential interactions of dietary fiber with concomitant therapies or other clinical conditions have been ruled out.

## 6. Overall Fiber Effect on Microbiota, Inflammation, and Intermediate Metabolism

There are many evidences that dietary fiber might be associated with decreased systemic and local inflammatory reactions and therefore with better clinical outcomes in acutely ill patients. Most of these effects seem to be mediated by the production of short chain fatty acids (SCFA), among which the most common are acetate, propionate, and butyrate. These molecules are the byproducts of the fermentation of complex indigestible carbohydrates by intestinal anaerobic microbiota. Fermentable fiber, including inulin-type fructans, pectins, resistant starches, and oligosaccharides, are the main precursors of SCFA; common sources in the diet include cereals, whole grains, raw fruits, vegetables and legumes [76,77]. Intestinal fermentation of fiber by *Bacteroidetes* spp. results mainly in the formation of acetate and propionate, while Firmicutes are responsible for the synthesis of butyrate [77]. SCFA exert a pivotal role in maintaining intestinal barrier integrity and in promoting growth and diversity of gut microbiota. While a variety of intact fibers characterize plant-sourced foods, fiber in functional foods and nutritional supplements may differ for origin (natural-extracted or purified, or synthetic) and most importantly diversity, as usually only one or a limited number of selected types of fibers are included [78]. As a result, comparing the effects of nutritional interventions based on different fiber supplementation to those of high-fiber diets is challenging, as well as assuming that the benefits of dietary fiber in natural plant-based foods documented by several studies apply also to supplemental fibers. In addition, it should be considered that intestinal microbiota production of SCFA depends not only on the type and amount of fermentable fiber in foods, but also on microbiota composition, diversity, amount of different taxa, and intestinal transit time.

From a local standpoint, SCFAs, and in particular, butyrate, improve the intestinal epithelial barrier in human colon cells by inducing a reduced transcription of the MUC2 gene and therefore determining a reduced production of mucine [79]. Butyrate and other SCFAs directly inhibit the NF-κβ signal in colonic cells and also activate the nuclear transcription factor PPARγ, which in turn counteracts the NF-κβ signaling pathway [79]. The NF-κβ signaling pathway modulates the transcription of genes that encode inducible inflammatory enzymes, proinflammatory cytokines, chemokines, and some acute phase proteins. SCFAs also provide fuel to colonocytes and regulate their proliferation and differentiation, increase colonic blood flow, reduce colonic pH, stimulate pancreatic secretions, promote sodium and water absorption, and possibly modulate gut motility. These evidences that have been described on molecular or tissue bases have been confirmed also in clinical studies, with intriguing results. For example, reduced intestinal permeability has been demonstrated in critically ill patients after eight days of fiber supplementation [80]. In the same study, a reduction of C-reactive protein associated with high fiber intake was observed.

From a systemic point of view, it has been observed in a mouse model of influenza that SCFAs dampens inflammatory reaction through multiple pathways [81]. SCFAs enhance bone marrow production of Ly6c-monocytes, which in turn lead to an enhanced production of macrophages with a lower ability to secrete the chemokine CXCL1 in the respiratory tract. A reduced production of CXCL1 results in a blunted recruitment of neutrophils in the lungs, thus restricting tissue damage during infective episodes. SCFAs also enhance CD8+ T cell effector function by improving cellular metabolism. Moreover, SCFAs can increase the number and function of several immune cells, including T regulatory, T helper 1, and Th17 effector cells, thus modulating excessive inflammation. The importance of SCFAs in the so-called gut-lung axis has emerged during the recent COVID-19 pandemic. High levels of SCFAs in the gut were associated with reduced lung damage as a result of viral infections [82]. From a systemic perspective, the proportion of colon-derived SCFAs transferred to the circulation is 36% for acetate, 9% for propionate, and 2% for butyrate [83]. SCFAs are not only involved in the regulation of intestinal barrier integrity of systemic inflammation and of immune responses, but also in the modulation of intermediate metabolism and of energy homeostasis by acting on adipose tissue, skeletal muscle, and liver [84]. The beneficial effects on glucose metabolism are mediated by several direct and indirect mechanisms. SCFAs stimulate GLP1 and PYY secretion, resulting in increased insulin and decreased glucagon production, improved insulin sensitivity, and enhanced gut motility [84,85]. In the liver, SCFAs attenuate glycolysis and gluconeogenesis while stimulating glycogen synthesis. Similar effects have been described in skeletal muscle, where activation of the AMP kinase (AMPK) transduction pathway results in increased expression of GLUT4, which improves muscle glucose uptake. There are also some effects of fiber on lipid metabolism: SCFAs activate fatty acid oxidation, while inhibiting de novo synthesis and lipolysis in the adipose tissue [86]. These combined effects result in a net reduction of plasma-free fatty acid levels. Finally, a reduction of plasma cholesterol concentrations as a result of dietary fiber intake has been described, both in rodents and in humans [86]. These results might be irrelevant in critically ill patients, but in the long term a reduction of glycated hemoglobin has been demonstrated in diabetic patients [87], and although only speculatively, a better glycemic control might be hypothesized also among acutely ill patients.

Dietary fiber may improve gastrointestinal function by shaping the composition of gut microbiota and by blunting the adverse effects of antibiotics on intestinal microorganisms [88]. Dietary fiber influences gut microbiota, increasing the fecal abundance of several beneficial species, reducing the Firmicutes/Bacteroidetes ratio and improving gut microbial diversity [89].

Aging is associated with significant changes in the intestinal microbiota which depend on several factors, including physical functioning, living conditions, dietary patterns, and concomitant medications [90]. Due to these multiple potential influences, a high interindividual variability in gut microbiota composition has been described in older adults [90]. However, one important recurrent feature in this population is the loss of diversity, which is prominent in those with frailty [90] who are also at high risk of adverse clinical outcomes [91]. In addition, acute diseases are associated with dramatic changes in intestinal microbiota also determined by concomitant drug administration (i.e., antibiotics) and changes in nutrition patterns during a hospital stay. The resultant dysbiosis is a trigger for several infectious complications, including sepsis [92]. The impact of both critical illness and advanced age on gut microbiota has been recently reported by Mesa et al. [93] Compared to younger patients, older adults admitted to intensive care units showed a higher abundance of pathogenic intestinal bacteria, including Bacteroides, Escherichia, and Shigella. Although the association with clinical outcomes is not reported, the described changes in intestinal microbiota suggest a link with clinical complications typical of older acutely ill patients, including length of hospital stay and mortality. Therefore, the use of dietary fiber may positively impact both gut microbiota composition and patient outcomes.

From a clinical standpoint, there is very little evidence on the association between fiber intake and clinical outcomes in acutely ill patients, often with mixed results. For example, administration of an enteral formula containing soluble partially hydrolyzed guar to a cohort of patients with severe sepsis did not modify the length of hospital stay and mortality risk as compared with standard feeding [94]. No difference in ICU mortality, length of hospital stay, or gastrointestinal complications between patients receiving a fiber or a non-fiber-containing enteral formal has been confirmed by a recent metanalysis [95]; however, the former group demonstrated a 39% lower risk of gastrointestinal complications [95]. Through a Bayesian analysis of randomized controlled trials conducted among critically ill adults, the supplementation of probiotics and prebiotics (i.e., symbiotics) has been demonstrated to be more effective than parenteral nutrition and probiotics and prebiotics alone in preventing urinary tract infection, hospital-acquired pneumonia, catheter-related bloodstream infection, and sepsis [96].

Besides outcomes related to metabolism and inflammation, fiber supplementation to enteral nutrition has been demonstrated to decrease the risk of diarrhea in many studies, reviews, and metanalyses [64,95,97,98,99,100,101]; additionally, other complications of enteral nutrition were reduced, such as regurgitation [97], vomiting [97], constipation [97,99], and abdominal distention [102]. A decrease of time to reach full enteral nutrition [64,97] was also shown, as well as a reduced duration of stay in the intensive care unit and in the hospital [97]. Finally, mortality was reported to be affected in one review [97].

Considering that among older hospitalized patients, mortality is higher in those who are more frail, the finding of a differential gut microbiota composition between frail versus nonfrail elderly subjects [100] who demonstrate higher abundance of SCFA producers has relevant implications. Although the impact of microbiota composition on clinical outcomes of frail older patients acutely ill remains to be fully elucidated, nutritional intervention based on dietary fiber may represent a strategy of treatment, since high fiber intake is associated with increased microbiota diversity in older subjects [103]. Concomitantly, tolerability and potential side effects of nutritional interventions with dietary fiber should be considered. In addition to potential interactions with drug absorption, early satiety as a result of the effect of the SCFA acetate, which can cross the blood brain barrier and suppress appetite, should be considered, particularly in malnourished older patients [104]. Gastrointestinal discomfort and bloating as results of intestinal fermentation may reduce the tolerability of high-fiber diets, while when fiber is included in dietary supplements, dislike of flavor or texture may limit compliance [78].

The overall effects of dietary fiber on older patients acutely ill are summarized in Table 1 and Figure 2.

## 7. Conclusions

Dietary fiber demonstrates several potential beneficial effects for both healthy and frail older patients (Figure 1). First, by modulating inflammaging it contributes to improved clinical outcomes in several aging-associated conditions. The effects on inflammaging are mainly mediated by improving aging-related gut dysbiosis, which contributes to the development of leaky gut and of low-grade chronic inflammation associated with several chronic diseases and frailty. Additionally, modulation of gut microbiota by fiber administration has an impact on metabolic health and insulin sensitivity, which is partly driven by the metabolic phenotype. A recent study conducted in a cohort of adult and older patients with wither liver- or muscle specific insulin resistance demonstrated that the latter group showed the highest benefit from a dietary intervention based on macronutrient manipulation, including high fiber intake [105]. These findings emphasize the complexity and heterogeneity of the interindividual metabolic and antiinflammatory responses to dietary fiber and justify the lack of data in the acute care setting. Moreover, in the general population, a recent metanalysis considering carbohydrate quality, glycemic index and load, and several clinical outcomes concluded that the available studies are inconclusive because they provide moderate or low-quality evidence. However, the contribution of lowering the glycemic index and glycemic load of dietary fiber appeared modest [106] when compared to its overall protective effects. These findings highlight the concept that dietary interventions based on fiber manipulation might exploit their beneficial effects if conducted with a view of precision nutrition. This is of special interest, since inadequate dietary fiber consumption has been reported for the older population [107]. However, the potential effects on the modulation of glucose metabolism and insulin resistance in the setting of those acutely ill remain so far undetermined.

## Figures and Tables

**Figure 1 nutrients-15-02365-f001:**
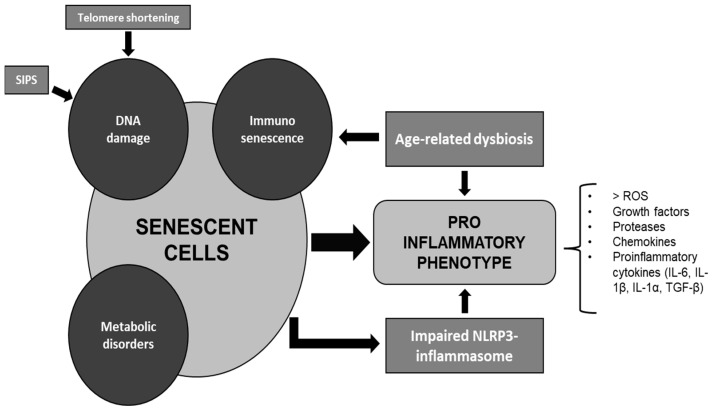
Cellular senescence induces a proinflammatory state. DNA damage (induced by SIPS and shortening of telomeres) immunosenescence, and metabolic disordes induce cellular senescence, which in turn leads to a proinflammatory state. Age-related dysbiosis and NLRP3 inflammasome impairment contribute to this mechanism.

**Figure 2 nutrients-15-02365-f002:**
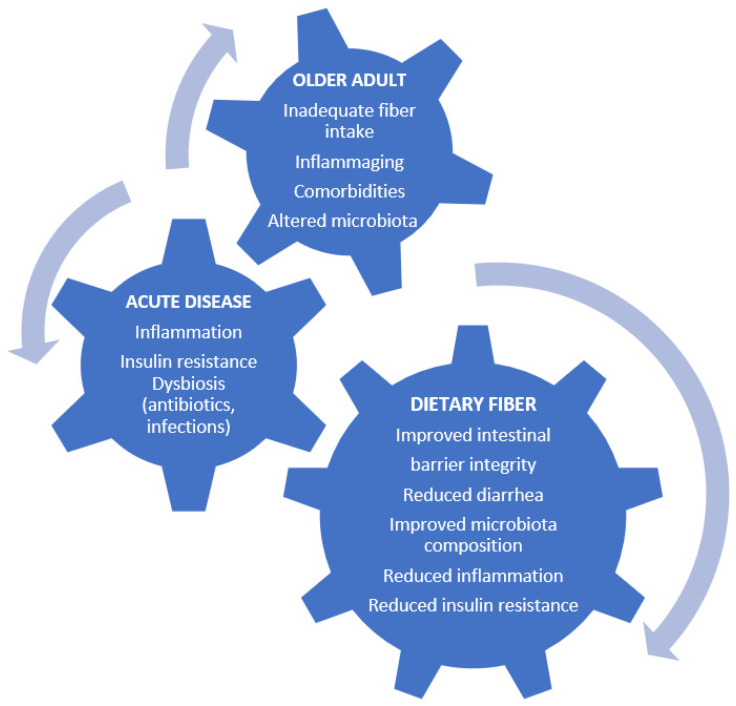
Potential effects of dietary fiber on metabolic and systemic derangements of acute conditions in older patients.

**Table 1 nutrients-15-02365-t001:** Metabolic, antiinflammatory, and intestinal effects of dietary fiber in older patients in the acute setting.

Effect	Results
Reduced inflammation	Reduced CRP in older adults acutely ill [57]
Improved insulin resistance	Lower insulin requirement in older neurocritically ill [57]
Diarrhea	Reduced in the acutely ill [57,76,77,78,79,80,81]

## Data Availability

Data sharing not applicable.

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
