# Peer review of "Impact of Dietary Fiber on Inflammation and Insulin Resistance in Older Patients: A Narrative Review"

_nutrients, 2023, doi:10.3390/nu15102365_

Round 1

Reviewer 1 Report

The authors have reviewed the mechanisms by which fiber may exert beneficial effects on metabolism in elderly patients.  Several issues should be considered

1. reference 85 is critical and should be summarized early in the paper

2. there should be more discussion of the effects of fiber on the microbiome and generation of SCFA and the mechanisms by which SCFA can modulate insulin sensitivity

3. can the authors provide information regarding the source of fiber and relative clinical benefits: for example can they compare fiber in foods vs dietary supplements.  Which types of foods are likely to be most effecttive?

4. please discuss tolerability and any possible negative effects of fiber in elderly patients 

suggest review by English- speaking colleague

Reviewer 2 Report

This review discussed the overall effect of fiber on gut microbiota, inflammation and intermediate metabolism, and summarized the evidences on the potential impact of dietary fibers in inflammation and insulin resistance during acute diseases with a special focus on older patients. The topic is interesting and innovative, which included the latest literature in the field. Furthermore, the paper has good logical writing. However, the following points should be tackled with before possible consideration for acceptance.

1. In the fifth point of the article, when discussing the relationship between dietary fiber and inflammation, some potential mechanisms of soluble dietary fiber in improving inflammation were described. However, these related studies were animal or in vitro studies, and there was no any evidence about human (especially about the elderly people). It would be better to add evidences from human studies regarding this point.

2. In the lines 327-330, it was mentioned that the effects on inflammaging are mainly mediated by improving aging-related gut dysbyosis. The dysbyosis should be dysbiosis.

3. There were many paragraphs in the second part (Mechanisms of chronic inflammation in the older adults) of the article . However, coherence was lacked between the paragraphs. Therefore, it is suggested to make some adjustment and improvement.

4. In the lines 242-244, it was described that "In older adults with acute diseases dietary fibers may induce abdominal distension and osmotic diarrhea, and possible interactions with drug absorption should be considered before their administration ". However, reference [63] described the factors that affect the absorption of levothyroxine in the human body, and there was no evidence that dietary fiber causes abdominal distension and osmotic diarrhea in older adults with acute diseases. In addition, should more attention be paid to the effect of dietary fiber on the absorption of routine medication for common chronic diseases in the elderly?

English language is fine.

Reviewer 3 Report

The manuscript entitled “Impact of dietary fiber on inflammation and insulin resistance in older patients: a narrative review” summarized the evidence for the influences of dietary fiber on inflammation and insulin resistance during acute diseases, focusing on older patients. Generally, this review work is of some significance for providing understandings about the roles of dietary fiber prevention and management of several chronic conditions associated with aging, including diabetes, neurodegenerative, cardiovascular diseases and cancers. But there are still some points need to be addressed.

1. The aims and the conclusions of the present review should be mentioned in the abstract part.

2. The authors used so many words to introduce the mechanisms of chronic inflammation, inflammaging and insulin resistance, inflammation in the older adult acutely ill in the part 2, part 3 and part 4, however, the authors just discussed the roles of dietary fiber on inflammation and insulin resistance in the part 5 and part 6. This seems to make the whole work deviate from the topic. Some improvements should be made.

3. I have read the whole manuscript, and know that in most of the paragraphs, the authors just have summarized the results from the previous studies. However, I would rather to know what could be exactly concluded from these previous studies from the authors’ point of view. These viewpoints might be more useful for the readers to know the progress of the research in the related areas.

4. Line 53-66. These short paragraphs can be incorporated in one paragraph. Other similar and short paragraphs also can be incorporated in one.

5. The figure 1 and figure 2 can be improved, they are not so informative.

6. The title of the present work is “Impact of dietary fiber on inflammation and insulin resistance in older patients: a narrative review”, how I wonder why the authors use so many words to carefully introduce the chronic inflammation, inflammaging, inflammation and insulin resistance, instead of focusing more on the topic to discuss the roles and the underlying regulation mechanisms of the dietary fiber in the inflammation and insulin resistance regulations?

7. The languages in the whole manuscript is generally good. But the authors are still suggested to make double check, for some sentences are not native English, or in a unusual way. E.g. Line 263, Line 278 (grammar mistake), Line 300, Line 285……

8. An abbreviations table can be added in the end of the conclusion part.

The languages in the whole manuscript is generally good. But the authors are still suggested to make double check, for some sentences are not native English, or in a unusual way. E.g. Line 263, Line 278 (grammar mistake), Line 300, Line 285……

An abbreviations table can be added in the end of the conclusion part.

Round 2

Reviewer 1 Report

the paper is much improved and more interesting

line 285: change "assumption" to "consumption"

line 285: change "assumption" to "consumption"